# What's in a database? Insights from a retrospective review of penguin necropsy records in Aotearoa New Zealand

Stefan Saverimuttu[1,2¤*], Stuart Hunter[3], Brett Gartrell[3], Kate McInnes[4], Kristin Warren[5], An Pas[1], James Chatterton[1], Lian Yeap[5], Bethany Jackson[2]

1 New Zealand Center for Conservation Medicine, Auckland Zoo, Auckland, New Zealand, 2 Centre for Biosecurity and One Health, Harry Butler Institute, Murdoch University, Perth, Australia, 3 Institute of Veterinary, Animal and Biomedical Sciences, Massey University, Wellington, New Zealand, 4 Department of Conservation/Te Papa Atawhai, Nelson, New Zealand, 5 Centre for Terrestrial Ecosystem Science and Sustainability, Harry Butler Institute, Murdoch University, Murdoch, Australia

¤ Current Address: Wildlife Health Services, Zoological Society of London, London, United Kingdom
* stefan.saverimuttu@zsl.org

## Abstract

Wildlife necropsy databases often provide data for morbidity and mortality studies of free-ranging species, with implicit relevance for conservation goals, as well as domestic animal and human health. Retrospective reviews are a common way to derive insights from such opportunistic data, despite the methodological difficulties of performing these analyses, alongside findings being prone to bias. This study reviews morbidity and mortality data from Sphenisciformes of Aotearoa New Zealand, using records extracted and manually refined from submissions to the national Wildbase Pathology Register. The review corroborates the broader consensus that hoiho (yellow eyed penguin, *Megadyptes antipodes*) are most commonly diagnosed with infectious/inflammatory disease (43.1%, 422/978 diagnoses), kororā (blue penguin, *Eudyptula minor*) with traumatic injuries (42.9%, 156/364 diagnoses), and emaciation being a common finding across both species (33.9%, 393/1463 diagnoses). Further, there are marked spatiotemporal trends in submissions, driven primarily by the affected species and the submitting organisations, highlighting the biases within such databases that must be factored into the application of results. Typographical errors, redundancies from synonymous terms, and missing data are captured as barriers to performing manual reviews of free-text data. Overall, this study highlights strengths and limitations of storage and review of wildlife necropsy data while providing insight into threats faced by the penguins of Aotearoa.

## Introduction

Effective conservation of free-living wildlife is predicated on three components: ensuring population health and viability, functionality within the ecosystem, and appropriate geographic representation [1]. All three are influenced by climate change and must account for changes to species phenology [2,3], meaning multidisciplinary expertise must be employed to derive an effective evidence-base for conservation. Active or passively acquired wildlife health data

**Data availability statement:** The data for this research is housed in a database (the Wildbase Pathology Register) that is subject to third-party IP arrangements and is contractually managed between a government body (Department of Conservation) represented by Dr Kate McInnes, and two Massey University project leads (both authors; Professor Stuart Hunter and Professor Brett Gartrell). As such it is these authors who manage any data requests to safeguard the confidentiality of all contributors and in turn the continuity of the database itself. The authors can be contacted using the following details: Brett Gartrell - b.gartrell@massey.ac.nz Kate McInnes - kmcinnes@doc.govt.nz Stuart Hunter - s.hunter@massey.ac.nz Access to the data will be granted for any legitimate scientific interests provided assurances can be given to safeguard the IP and confidentiality of all 3rd party database collaborators. Although the authors cannot make their study's data publicly available at the time of publication, all authors commit to make the data underlying the findings described in this study fully available without restriction to those who request the data, in compliance with the PLOS Data Availability policy. For data sets involving personally identifiable information or other sensitive data, data sharing is contingent on the data being handled appropriately by the data requester and in accordance with all applicable local requirements All geographic boundary datasets used in this study are publicly available. The coastline and island polygons dataset is available from Land Information New Zealand (LINZ) (https://data.linz.govt.nz/layer/51153-nz-coastlines-and-islands-polygons-topo-150k/), and territorial authority boundaries are available from Stats NZ (https://datafinder.stats.govt.nz/layer/111194-territorial-authority-2023-generalised/).

**Funding:** The research was funded by a partner scholarship between Auckland Zoo and Murdoch University, as part of the Conservation Medicine Residency program. The funders had no role in study design, data collection and analysis, decision to publish, or preparation of the manuscript.

**Competing interests:** The authors have declared that no competing interests exist.

informs these components of conservation, with implications for domestic animal and human health [4]. Although wildlife necropsy databases comprise a key source of wildlife health data [5,6], their utility in species conservation relies on linking data acquisition, representation of the target population, data analysis, and delivery of findings to those that manage species and ecosystems. Thus, to gain contemporary insights from necropsy databases, comprehensive reviews would ideally be undertaken routinely, with results interpreted within the ecological context of the species involved, and the inherent biases of the dataset.

In the field of wildlife health, morbidity and mortality studies are often based on sporadic review of opportunistically obtained wildlife necropsy data or 'convenience sampling' [7–10]. This contrasts with the more purposeful and systematic/randomised collection of similar data in production animal health, often analysed and communicated in real-time. This time sensitive data analysis in production animals is likely driven by their obvious economic value, as well as the relative ease of applying robust epidemiological sample sizes and surveillance designs to obtain samples representative of the target population, as seen in the broader literature [11–13]. Such data can then be used to identify new or changing disease patterns, or determine the success of any interventions, improving productivity and animal welfare, as well as access to export markets [14]. In contrast, the goals of wildlife health monitoring, such as through necropsy databases, centre on informing conservation policy, species management, and identification of emerging infectious diseases which may pose a threat to wildlife, domestic animal or human health [15,16]. The perceived economic value of healthy ecosystems, inclusive of wildlife, is often indirect [17], making the cost-benefit of real-time health data analysis less tangible. In either case, such health monitoring comes with the logistical and fiscal challenges of collecting adequate and statistically meaningful sample material, as well as sourcing the appropriate equipment and expertise for processing [18]. Timely reporting of findings and inclusion in policy or management will be augmented by having established systems, funding, and personnel to manage data acquisition, analysis, and interpretation from the outset of any animal health data collection programs.

The limitations of data interpretation from wildlife necropsy databases are largely conserved across publications, with selection/admission bias, misclassification bias, type II errors (owing to small sample sizes), and confounding, all common issues [15,19–21]. Passively acquired wildlife necropsy data are likely akin to that from other omnibus surveillance programs, fundamentally lacking external validity due to a plethora of systematic errors (bias) [22–24]. Submissions are influenced by the visibility or location of a species, alongside research and conservation priorities. As such, species with a low threat status, remote geographic range, or cryptic lifestyle may be under-represented [19]. The opportunistic nature of sample collection consistently leads to recognised biases in the species demographics and even diagnoses obtained [7,10]. The identification of where such biases may apply cannot rely solely on careful data analysis. Potential insights from opportunistic wildlife health data must be filtered through the lens of a multidisciplinary team to recognise ecological factors which may drive pathologies observed. More importantly the integration of health and ecological priorities likely better utilises resources while leading to more holistic management solutions [25]. Thus, health issues identified from necropsy databases will benefit from targeted and systematic investigations to derive a clearer epidemiological picture, and resolve apparent biases in pathology seen [10,20]. However, the very structure of a necropsy database can pose significant challenges to effective review, with issues ranging from record completeness to consistency of reporting of key metadata and risk factors of interest [9,26,27]. Consequently, while passively acquired wildlife necropsy databases serve as valuable sources of information, findings derived from them must be cautiously applied to conservation policy and management decisions.

New Zealand's national catalogue of wildlife necropsies, The Wildbase Pathology Register, is a pertinent example of an opportunistically compiled wildlife necropsy database. Through an agreement between *Te Papa Atawhai* (New Zealand government Department of Conservation) and Massey University, threatened species are submitted from around the country for necropsy at Wildbase Pathology. As with similar repositories, the Wildbase Pathology Register holds information of value to wildlife conservation endeavours, such as the management of threats to New Zealand's iconic *Sphenisciformes.* The mainland and neighbouring islands of New Zealand provide breeding grounds for six of the world's eighteen extant penguin species [28]. These animals considered *taonga* ('living treasure'), play important roles in the region's ecology, culture, and economy [29]. Despite this societal significance, the kororā (blue penguin, *Eudyptula minor)* is New Zealand's only penguin species listed internationally as 'Least Concern,' with all others listed as at least 'threatened' [30]. Threats faced by New Zealand *Sphenisciformes* appear largely anthropogenic in nature [31] including industry [32], introduced pests [33,34], ecotourism [35,36], and climate change [29]. The iconic and endemic hoiho (yellow eyed penguin, *Megadyptes antipodes*) is one of five penguin species globally for which infectious disease has been recognised as a specific conservation concern [28,37], reflecting two large scale mortality events with no specific aetiology identified [38,39], the enigmatic syndrome of diphtheritic stomatitis [37] and the emerging threat of avian malaria [40]. Details of necropsies performed in the process of these investigations (and others) have been stored in the Wildbase Pathology Register, making this a potential source of information to contribute to the conservation of these taonga.

In this study, we aimed to review *Sphenisciformes* necropsy records stored in the Wildbase Pathology Register, as a case study to highlight opportunities and limitations for such wildlife health databases to contribute to wildlife conservation in New Zealand and globally. Thus, we performed a traditional descriptive review of the demographic and epidemiological features of the dataset, including an evaluation of the data's structural integrity. With this, we intended to identify (i) predicted limitations of demographic representation across species, (ii) co-variance between key signalment variables, and (iii) sources of bias and confounding. Finally, we quantified clinicopathologic features in accessions to the database. Using this review, we seek to establish priorities for future development and use of such important sources of wildlife health information, to streamline the process and transparency from data acquisition to conservation policy and management.

## Materials and methods

Data for this retrospective review was derived from the Wildbase Pathology Register and included post-hatch, whole carcass accessions from February 1986 to July 2020. Deceased wildlife from all Aotearoa New Zealand (including surrounding islands), are submitted by government rangers and members of the public, alongside commercial and scientific ventures, for necropsy at Wildbase Pathology. These submissions are predominately of individuals of species classified as "threatened" according to the 'New Zealand Threat Classification System' [41], and a small selection from any mortality event of 3 or more animals. In addition, collaborating institutions may perform the gross necropsy of suitable wildlife specimens and submit a report along with formalin fixed tissues for histopathology. Necropsy protocols are recorded on a standard form, individually identified in a cloud-hosted database. All data (signalment, location, history, gross and histopathological findings, and diagnostic results) are recorded in individual fields however most responses are not standardised nor mandated; thus, the majority are free-text.

All wild penguin records were individually reviewed within the online database. Data ancillary to the pathological descriptions (age classification, sex, submitting organisation, species, and body condition score) were entered into commercial spreadsheet software ('Google

sheets'). Due to the free-text nature of these data fields, some interpretation of the recorded information was performed by the primary author to standardise input to the spreadsheet (see Table 1).

Gross and histopathological findings were transcribed from relevant sections of each report by organ system, alongside ancillary testing results such as molecular and microbial investigations. Morphological or etiological diagnoses, when made explicit, were transcribed as reported. All recorded diagnoses were evaluated against a list of those found previously to guard against synonymous term usage unnecessarily complicating the dataset (e.g., diagnoses reported as 'starvation', 'emaciation', or 'malnutrition' were all recorded as 'emaciation' if evidence of clinically significant low body condition was reported). Diagnoses were recorded as specifically stated within each report with more, or less specific variants of diagnoses recorded as separate entities (e.g., 'pneumonia' and 'bronchopneumonia' recorded as separate diagnoses). The proximate cause of death was not recorded, as this was not clearly identified in most records. Therefore, all diagnoses were considered equally important and were not ranked. No *a priori* limit was placed on diagnoses allowed per record during the data collation process. Once completed the spreadsheet was imported into the 'R' statistical software platform through the 'R Studio' interface.

Initial analyses consisted of data exploration and descriptive statistics of signalment and non-clinical ancillary data. Cramer's V statistics were used to infer covariation between categorical variables (age class, year, sex, species, body condition) within the dataset, with significance set at $P<0.05$. Given the vast majority of accessions were hoiho and kororā, further analyses were restricted to these two species. To investigate associations between sex, age class, month, and year, with being either a hoiho or kororā submission, we used logistic regression analysis including odds ratios, 95% confidence intervals, and p-values. For the category of year, we restricted the dataset to the years 2001–2019 inclusive, owing to low total accessions (<10 per year) prior to 2001, and 2020 having data for only half of the year.

Submitter and location data were extracted from the relevant sections of each necropsy report and tabulated. Redundancies in reporting of these variables were manually resolved and descriptive statistics calculated. All location related fields of each report (when present) were used to approximate the location from which each specimen had been submitted. These data were combined with two spatial datasets for geographic boundaries in New Zealand to generate a map comparing submission numbers by location to human population density estimates in ArcGISPro. Coastline and island polygons were sourced from Land Information New Zealand [42]. Territorial authority boundaries were obtained from Stats New Zealand [43]. Both datasets were downloaded in shapefile format and licensed under Creative Commons Attribution 4.0 International.

**Table 1. Ancillary data of whole carcass, post-hatch *Sphenisciformes* accessions stored in the Wildbase Pathology Register to July 2020 which required interpretation to standardise spreadsheet input during necropsy review.**

| Variable | Native Levels | Interpreted Levels | Explanation |
|---|---|---|---|
| *Age classification* | 4 | 3 | Native levels of 'Juvenile' and 'Subadult' treated as synonymous. |
| *Sex* | 4 | 3 | Some records denoted as 'Male/Female', these reallocated as 'Unknown'. |
| *Submitting Organisation* | 130 | 46 | Typographical errors, alternative names, and alternative spellings amalgamated. |
| *Species* | 14 | 9 | Species consolidated to account for spelling variations, typography, and synonyms at the subspecies level according to IUCN listed taxonomy |
| *Body Condition Score* | Variable | 5 | Written descriptors were interpreted into a 1–5 scale. For data originally reported on a 1–9 system, the values were scaled to a 1–5 system by dividing by 1.8 (the ratio of 9:5) and rounding to the nearest integer. |

A frequency table of clinical diagnoses was extracted from the main spreadsheet whilst maintaining species associations. The clinical diagnoses listed were manually inspected to correct for redundant synonyms or typographical errors prior to analysis. Each unique diagnosis in this list was then classified according to the 'DAMNITV' etiological classification system (degenerative, anomalous, metabolic, nutritional and neoplastic, inflammatory and infectious, traumatic, vascular) as is widely used in the broader veterinary literature [44–46]. These classifications were then mapped back to the individual accessions where each diagnosis was recorded. Data exploration consisting of descriptive statistics, and graphical visualisations were derived for individual diagnoses (only including those observed in ten or more instances) and each DAMNITV diagnosis class. Analyses were performed on the dataset with all species included, followed by the hoiho and kororā data separately.

## Results

A total of 1300 individual *Sphenisciformes* necropsy reports were examined from the Wildbase Pathology Register, spanning 34 years, with the earliest accession from 1986 and the latest accession from July 2020.

### Signalment variables (species, sex, age classification)

Seven penguin taxa were represented in the database; hoiho were the most populous (67.8%, 882/1300), followed by kororā (25.2%, 328/1300). The remaining individual records were comprised of the; tawaki (Fiordland crested penguin, *Eudyptes pachyrhynchus*) (4.9%, 64/1300), Snares crested penguin (*Eudyptes robustus*) (0.8%, 11/1300), erect crested penguin (*Eudyptes sclateri*) (0.8%, 10/1300), royal penguin (*Eudyptes schlegeli*) (0.3%, 4/1300), and a single (0.08%, 1/1300) eastern rockhopper penguin (*Eudyptes filholi*).

Sex was specified in 76.2% (990/1300) of records, with males and females close to evenly represented (M:36.0%, F: 40.2%), and the remaining 23.8% designated as 'unknown' or sex not recorded. Age classifications of animals were specified in 93.2% of records (1211/1300), being classed as either 'juvenile' (38.5%, 500/1300), 'adult' (35.5%, 462/1300), or 'neonate' (19.2%, 249/1300), with no age classification determined for 6.8% of accessions (89/1300). Body condition score was specified on a one to five scale for 87.5% of records (1138/1300), with a distribution as follows: 1/5 (38.3%, 498/1300), 2/5 (14.5%, 188/1300), 3/5 (20.7%, 269/1300), 4/5 (13.8%, 179/1300), 5/5 (0.3%, 4/1300). Body condition score was not indicated in 12.5% of records (162/1300).

Pairwise comparisons of species, sex, body condition score and age classification across the entire dataset revealed statistically significant, but low-level covariation between species-body condition score, species-age classification, and age classification-body condition score (Table 2). The

**Table 2. Cramer's V statistics for pairwise comparison of signalment variables for whole carcass, post-hatch *Sphenisciformes* accessions stored in the Wildbase Pathology Register from to July 2020.**

| Variable pair | Cramer's V | p-value | degrees of freedom |
|---|---|---|---|
| *species-body condition score* | 0.110 | 0.002 | 28 |
| *species-sex* | 0.116 | 0.066 | 7 |
| *species-age classification* | 0.227 | <0.001 | 14 |
| *sex-age classification* | 0.086 | 0.031 | 2 |
| *sex-body condition score* | 0.045 | 0.767 | 4 |
| *age classification-body condition score* | 0.226 | <0.001 | 8 |

significance and magnitude of these relationships were unchanged with examination of covariation between hoiho and kororā only.

## Annual and seasonal trends

Year and month were recorded for 96.3% (1252/1300) of accessions. Total accessions by year ranged from 1–129 (mean 39.1, median 23.5), with an approximately two-fold increase in accessions from the previous year in 2005 and again in 2008 (Fig 1).

Penguin accessions fluctuated monthly, with most submissions during the November to April period (74.2%, 964/1300) (Fig 2).

## Comparing hoiho and kororā

Considering species influences, hoiho and kororā dominated the accessions overall, with hoiho driving the majority of yearly, monthly and age class associations (Table 3).

In the 19-year period from 2001–2019, hoiho were significantly more likely to be accessions than kororā in 53% (10/19) of the years, with 11.5% (101/882) of all hoiho submissions occurring in one year (2019). Across the whole dataset, hoiho submissions were significantly more likely than kororā in the months of January, February, March, April, May, and November. Hoiho submissions demonstrated greater monthly variation, peaking in November at

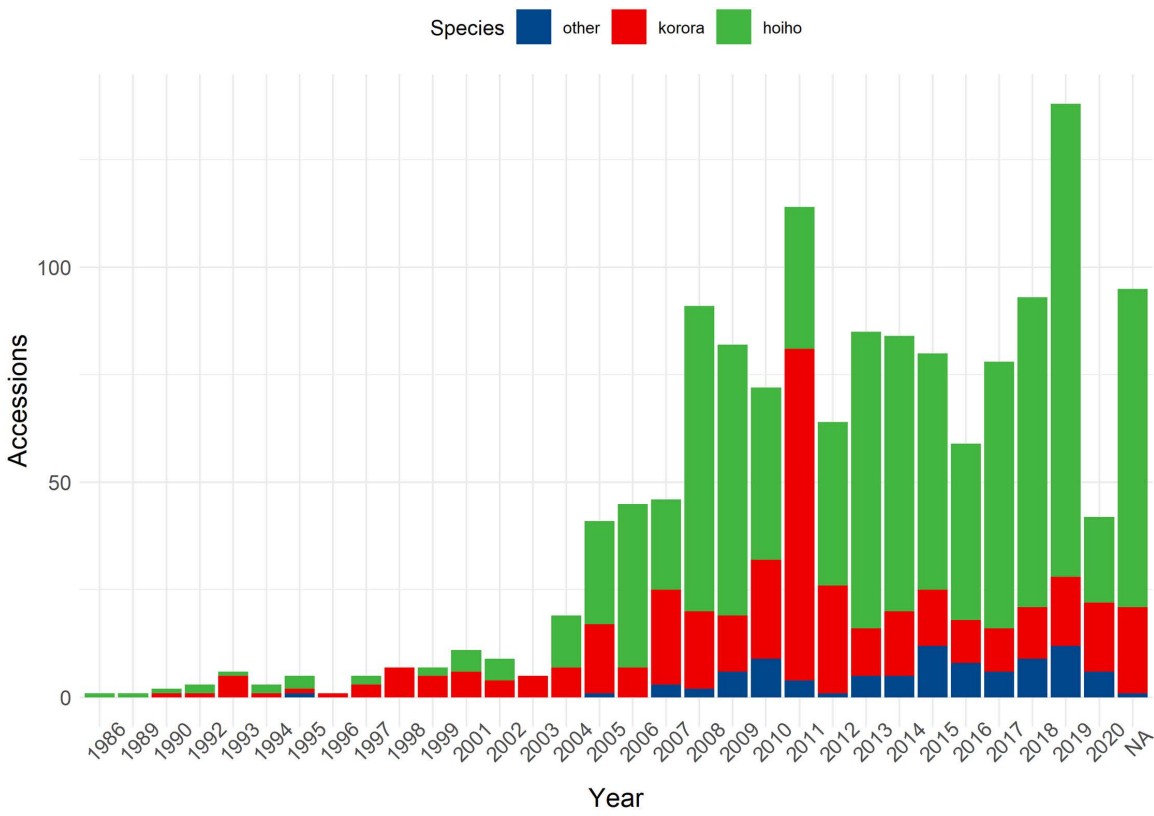

**Fig 1. Count of whole carcass, post-hatch *Sphenisciformes* accessions stored in the Wildbase Pathology Register to July 2020.** Total count of penguin accessions (y-axis) across years (x-axis) of the Wildbase Pathology Register of New Zealand. Species are represented by colours as per the figure key, with kororā (blue penguin, *Eudyptula minor*) and hoiho (yellow eyed penguin, *Megadyptes antipodes*) specifically represented as the most populous accessions. 'NA' on the x-axis represents records for which no year was recorded. 2020 shows relatively few accessions as the study period ends in July of this year.

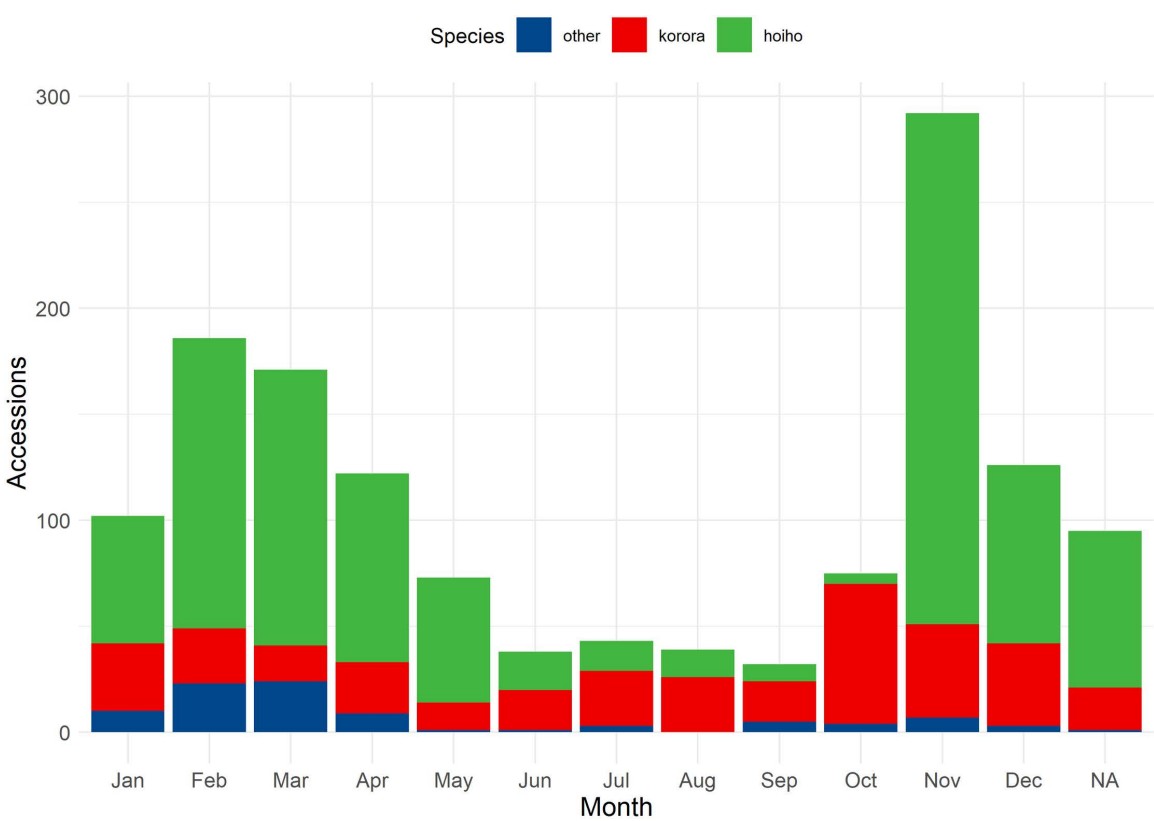

**Fig 2. Species distribution over all months summed across years, of whole carcass, post-hatch *Sphenisciformes* accessions stored in the Wildbase Pathology Register to July 2020.** Total number of penguin accessions (y-axis) entered in each month (x-axis) summed across all years of the Wildbase Pathology Register of New Zealand. Species are represented by colours as per the figure key with kororā (blue penguin, *Eudyptula minor*) and hoiho (yellow eyed penguin, *Megdyptes antipodes*) specifically represented as the most populous accessions. 'NA' on the x-axis represents records for which no month was recorded.

26.2% (231/882) of all accessions, with the majority (72.1%, 636/882) occurring from late Spring to early Autumn (November to March). Across the 34-year study period, 73.5% (172/234) of the November hoiho accessions occurred in six individual years (2006, 2008, 2010, 2013, 2017, 2019). Further, 74.8% (172/230) of hoiho accessions in November across the entire study period were from neonates, an age class that was significantly associated with being a hoiho (OR: 25.1, 95%CI: 11.21–71.66, p<0.0001).

Conversely, kororā submissions had less interannual and monthly variation, with 2011 comprising 15.5% (74/328) of all accessions, largely due to a spike in October of that year (51 accessions), compared to a mean of 0.58 per month of October for the years from 2001 to 2019, excluding 2011). Thus, in 2011 the odds of an accession being a kororā were significantly higher than a hoiho, compared to the years 2001–2019 (OR: 7.59, 95%CI: 4.89–11.77, p<0.0001). Kororā were significantly more likely than hoiho to be submitted for necropsy in July, August, September, and October.

## Submitter and location data

The submitting organisation was recorded in 98.8% (1285/1300) of records, with similar levels of data completeness for pathologist (97.8%, 1272/1300) and submitting individual (94.6%, 1230/1300). Once typographical errors and alternate names were accounted for in submitting

**Table 3.** Summary of significant variables associated with being a hoiho submission (versus a kororā submission) to the Massey Pathology Register. Odds ratios, 95%CI, and p-values from logistic regression models represent all years for age class and month) and are restricted to the years 2001 to 2019.

| Variable | N | Odds Ratio | 95% CI | P-value |
|---|---|---|---|---|
| *Age class* | | | | |
| *Adult* | 462 | (ref) | (ref) | (ref) |
| *Juvenile* | 500 | 1.72 | 1.59 - 2.38 | <0.001 |
| *Neonate* | 249 | 25.1 | 11.21 - 71.66 | <0.001 |
| *Year* | | | | |
| *2006* | 42 | 8.88 | 2.02 -43.53 | 0.004 |
| *2008* | 87 | 6.09 | 1.63- 23.93 | 0.007 |
| *2009* | 83 | 5.40 | 1.44- 21.28 | 0.012 |
| *2013* | 81 | 11.83 | 2.91 - 52.05 | <0.001 |
| *2014* | 80 | 5.23 | 1.39 - 20.62 | 0.014 |
| *2015* | 79 | 5.50 | 1.44 - 22.14 | 0.012 |
| *2016* | 57 | 6.15 | 1.52 - 26.61 | 0.011 |
| *2017* | 74 | 9.00 | 2.25 - 38.59 | 0.002 |
| *2018* | 91 | 7.75 | 2.02 - 31.38 | 0.003 |
| *2019* | 129 | 7.58 | 2.06 - 29.22 | 0.002 |
| *Month* | | | | |
| *January* | 96 | (ref) | (ref) | (ref) |
| *February* | 185 | 2.62 | 1.42 - 4.85 | 0.002 |
| *March* | 167 | 3.97 | 2.03 - 8.03 | <0.001 |
| *April* | 118 | 1.98 | 1.04 - 3.81 | 0.039 |
| *May* | 71 | 2.68 | 1.25 - 6.08 | 0.013 |
| *July* | 41 | 0.29 | 0.13 - 0.64 | 0.002 |
| *August* | 39 | 0.25 | 0.11 - 0.55 | <0.001 |
| *September* | 30 | 0.24 | 0.09 - 0.59 | 0.003 |
| *October* | 72 | 0.04 | 0.01 - 0.10 | <0.001 |
| *November* | 278 | 2.89 | 1.65 - 5.04 | <0.001 |

organisation, there were 43 unique submitting organisations identified, with the 'Department of Conservation' responsible for most individual records (63.2%, 821/1300).

The Wildbase Pathology Register provides four fields through which location data may be entered (Table 4). The most complete was 'Location Details' (82.4%, 1071/1300) denoting the local name for the area in which a specimen was found (e.g., a beach name). Both 'Location Type' (intended to describe if the animal died in the wild or captivity) and 'Location' (intended

**Table 4.** Ancillary submission data completeness of whole carcass, post-hatch *Sphenisciformes* accessions stored in the Wildbase Pathology Register to July 2020.

| Variable | Fields complete (%) | Description |
|---|---|---|
| *Location* | 20.4% | Descriptor for the local environment around the specimen (e.g., coastline, zoological park) |
| *Location Type* | 20.4% | Denotes if the animal died in the wild or in captivity |
| *City* | 92.5 | Nearest city to which the specimen was found |
| *Location Details* | 82.4% | Local name of the area in which the specimen was found |
| *Pathologists* | 97.8% | Name(s) of the pathologist(s) who performed the necropsy |
| *Organisation* | 98.8% | Name of the organisation submitting the specimen |
| *Submitter* | 94.6% | The individual submitting the specimen for necropsy |

as a descriptor for the local environment of the specimen, e.g., coastline, forest, or veterinary clinic) were poorly completed (both 20.4%, 265/1300). The field, 'City' representing the nearest city to which the specimen was found, was filled out in 92.5% of records (1202/1300).

A total of 1253 records had location data that facilitated mapping of submissions (Fig 3). The majority of submissions with location data (56.6%, 736/1300) were from Dunedin in the Otago Regional Council Area. Submissions were primarily aligned to coastal areas, with inland locations also represented. The majority of inland submissions were from Palmerston North (n=92), where Massey University and the Wildbase Pathology Service are located. The number of individual submissions did not correlate spatially with the projected human population census data by territory for 2023 (Fig 3), where the largest submissions would be expected in the territories housing the main cities of Auckland, Wellington (capital) and Christchurch.

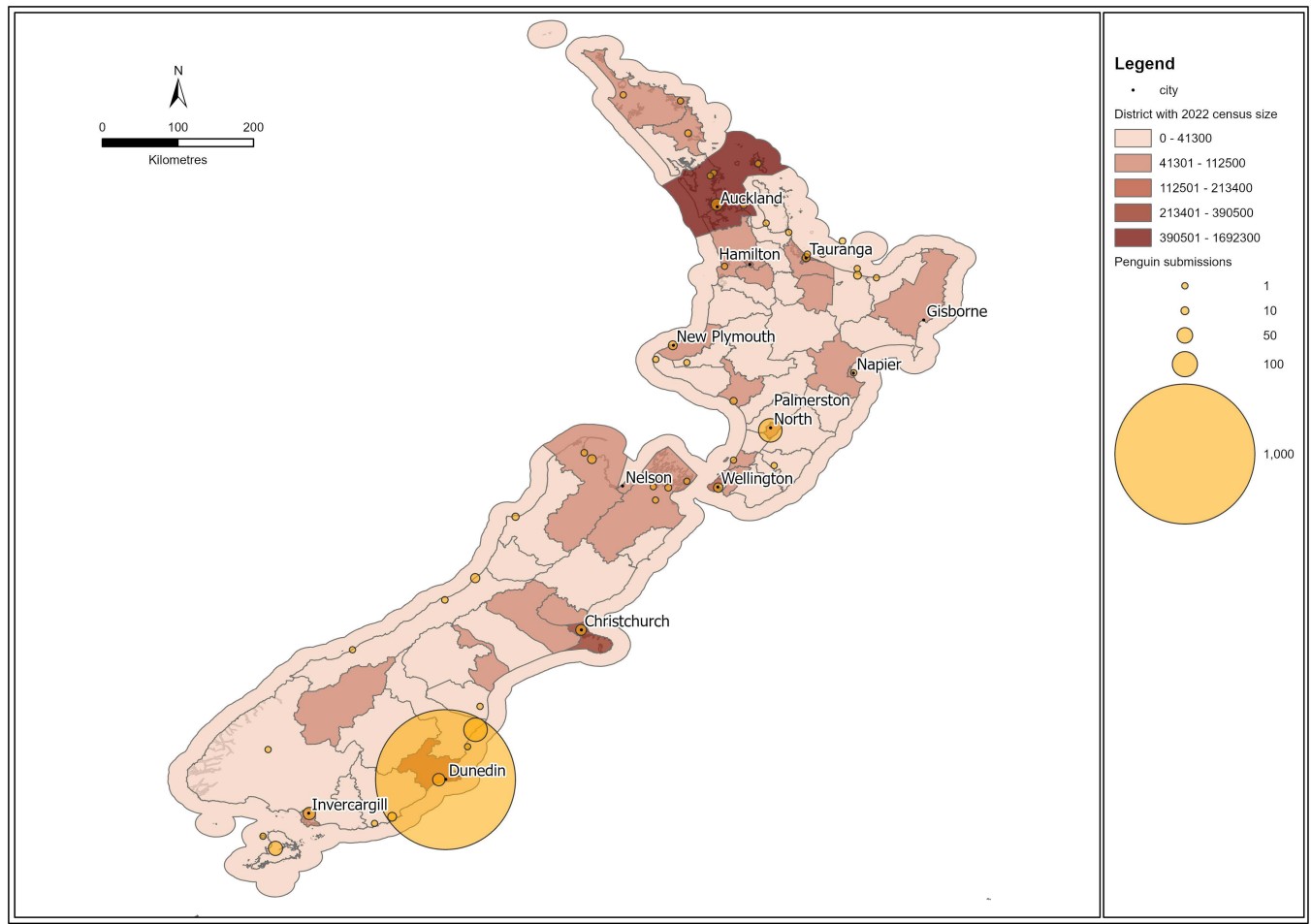

**Fig 3. Map of whole carcass, post-hatch *Sphenisciformes* accessions stored in the Wildbase Pathology Register to July 2020 and projected population densities across Aotearoa New Zealand.** Made in ArcGISPro the map shows submissions of *Sphenisciformes* (yellow circles) centred over the cities which are most closely associated with the submissions as recorded in the Wildbase Pathology Register. Brown shading indicates relative human population densities of the official New Zealand territorial boundaries as per the figure legend. Population data is drawn from projected population data for 2023 [47] from the New Zealand government. Coastline and island polygons were obtained from Land Information New Zealand [42]. Territorial authority boundaries were sourced from Stats NZ [43]. Both datasets are licensed under Creative Commons Attribution 4.0 International.

## Diagnoses

A total of 1463 discrete diagnoses were assigned to the 1300 individual records in the Wildbase Pathology Register (see Appendix 1 for complete list with classifications), with each record being assigned between 1–4 discrete diagnoses. There were 135 unique diagnoses, with only 19 occurring in 10 or more instances, and 79 diagnoses (58.5%) being derived only once. Across all species emaciation was the most frequent diagnosis (33.9%, 393/1463) and trauma (excluding predation) the second most common (17.6%, 258/1463) (Fig 4).

With individual diagnoses classified according to the DAMNITV system, across all species infectious and inflammatory diagnoses were the most frequent (35.7%, 523/1463), with traumatic (28.4%, 415/1463), and nutritional and neoplastic diagnoses (26.9%, 394/1463) following. Filtering data to observe only hoiho, the same pattern is seen, infectious and inflammatory diagnoses being the most frequent (43.1%, 422/978), with trauma (23.4%, 229/978), and nutritional/neoplastic diagnoses (23.2%, 227/978) in similar proportions. However, traumatic diagnoses, make up the largest proportion (42.9%, 156/364) of all diagnoses in kororā (Fig 5).

Carcass decomposition was explicitly noted as a limiting factor in the ability to derive diagnoses for 11.7% (152/1300) of necropsy records.

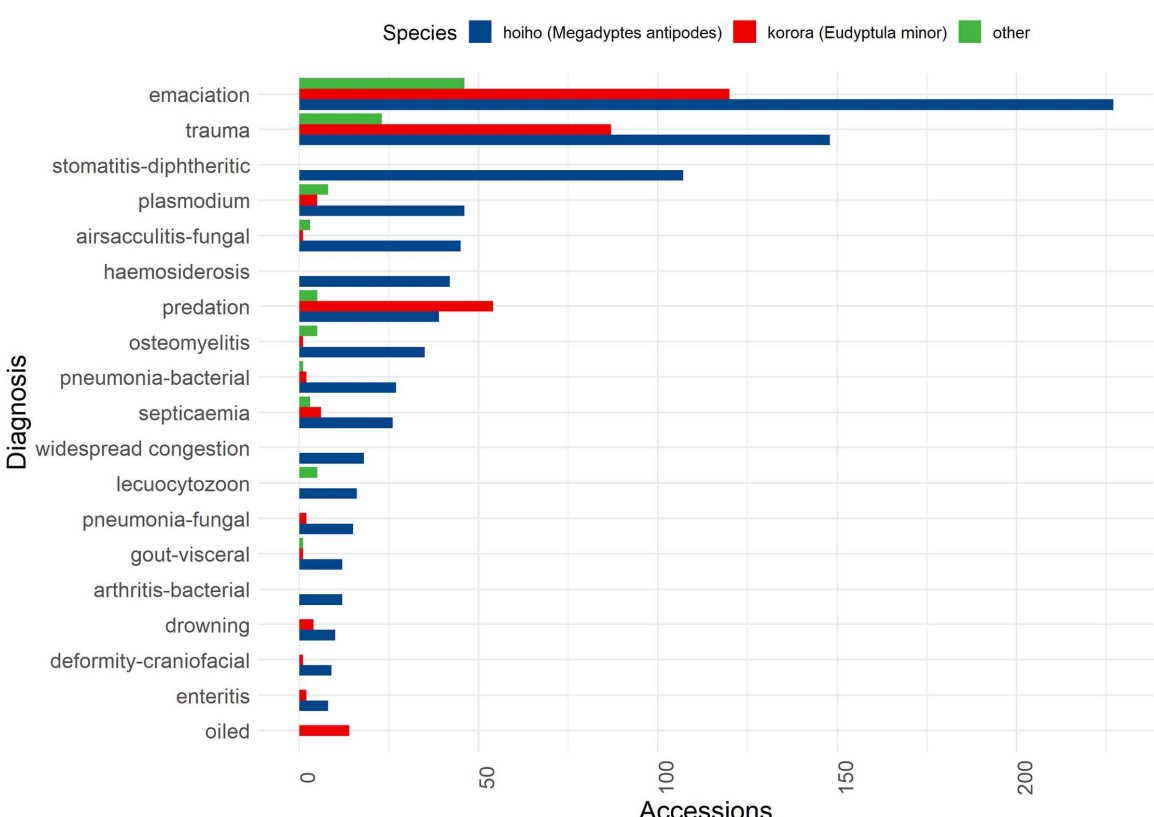

**Fig 4. Species distribution of all diagnoses encountered over 10 times in a review of whole carcass, post-hatch *Sphenisciformes* accessions stored in the Wildbase Pathology Register to July 2020.** Total number of accessions (x-axis) for which each diagnosis (x-axis) was assigned. Colour indicates if the bar represents hoiho (yellow eyed penguin, *Megadyptes antipodes*), kororā (blue penguin, *Eudyptula minor*), or other species as per the figure key. Individual findings which were assigned to fewer than ten accessions are not included.

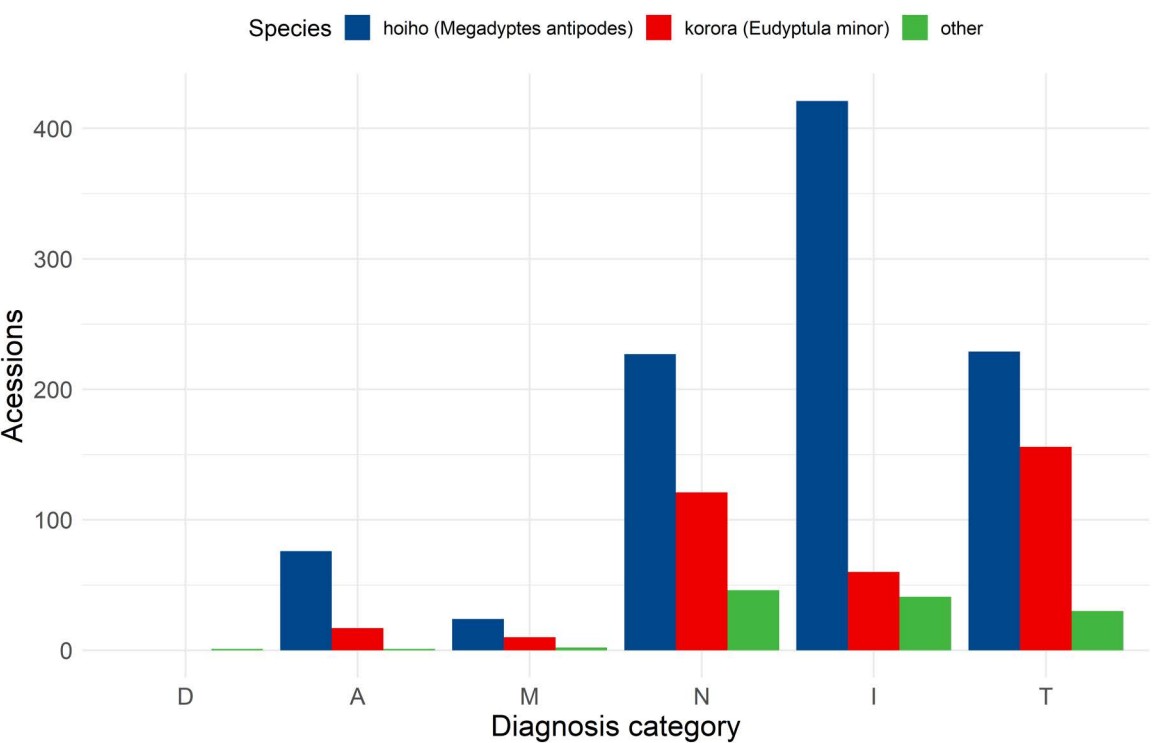

**Fig 5. DAMNITV grouping of diagnoses found in a review of whole carcass, post-hatch *Sphenisciformes* accessions stored in the Wildbase Pathology Register to July 2020.** Frequency of diagnoses (y-axis) grouped by category according to the DAMNITV mnemonic (x-axis). Letters in the mnemonic indicate broad categories of diagnoses as: D=Degenerative, A=Anomalous, I=Infectious and Inflammatory, M=Metabolic, N=Nutritional and Neoplastic, and T=Traumatic. Colour indicates if the bar is representative of hoiho (yellow eyed penguin, *Megadyptes antipodes*), kororā (blue penguin, *Eudyptula minor*), or other species as defined in the figure key.

## Discussion

In this review, we describe spatial and temporal trends in submission data for penguins in the Wildbase Pathology Register, noting the influence of variables such as year, month, age, species, and location of submission on the dataset. The species-based asymmetry in the database, particularly from 2005 onwards, means that crude trends and associations must be considered with this bias towards hoiho accessions in mind. The analysis of submitter and location data presented here suggests that a multiplicity of factors my drive biases in similar databases. These factors include submitting institution, research funding and field focus, stochastic, environmental, or anthropogenic disasters, and species visibility all of which have implications for how results are incorporated into management or research priorities. The proportional dominance of emaciation and traumatic injury across all species corroborates published findings of degrading diet quality and environmental conditions for penguins in New Zealand [2,29,48,49]. While the comparatively high proportion of infectious and inflammatory disease in hoiho may be suggestive of an interplay between traumatic injury, poor nutritional status, and infectious and inflammatory disease processes, the directionality of which is worth ongoing investigation through focused research, though is likely multidirectional [50]. The process of compiling this review highlighted the need to standardise the input of epidemiological and signalment data to increase the ease and value of future review processes [51].

Spatiotemporal species trends in the database suggest multiple factors influence the likelihood of submissions, primarily research priorities and intensity, often aligned to threat

status or public perception of a species, as well as random events. Collectively these factors contribute to funding and visibility which can offset the general drivers for observing mortalities in wildlife. Despite hoiho being more cryptic and distributed in less densely populated regions of New Zealand than the gregarious and coastal foraging kororā [52,53], substantially more hoiho are submitted overall due to intense monitoring of the mainland population for research and conservation interests, particularly in the summer breeding season where neonates form the majority of the dataset. However, there are no submissions from the subantarctic population stronghold of hoiho, likely due to the logistics of monitoring in this location, thus the database represents only one of two genetically distinct populations of this species [54].

The New Zealand Threat Classification System status of kororā is 'At Risk, Declining' [41] and therefore the threshold for submissions is generally mortality events of three or more birds, rather than individuals, which influences the nature of health events that may drive submissions. The spike in kororā submissions in October of 2011, aligns with the highly publicised and intensively managed Rena oil spill [55], again supporting the impact of individual events on the database. In comparison, the other five penguin species contribute minimally, likely reflecting their lower relative abundance on mainland New Zealand [56]. Although a similar pattern might apply to hoiho, the data suggests this is counteracted by significant public and conservation interest in the species. Further, spatial data presented in Fig 3 shows that density of submissions does not align with human population density. Rather, the coastal distribution of most submissions, clustering in the Otago Regional council area, and 'The Department of Conservation' being the organisation responsible for the majority of specimen collection (63.2%, 821/1300) supports the hypothesis that local research and conservation intensity has been a key driver for specimen collection. We propose that clear articulation of the goals and objectives of necropsy databases can help determine if active management of submissions (e.g., spatial, and temporal species limits or targets) will benefit analyses and interpretation, versus a truly ad hoc system which may detect outbreaks but risks domination by factors that introduce substantial bias.

Given the dominance of hoiho and kororā in the database, it is no surprise they drive seasonal and interannual trends. Submissions of hoiho decrease markedly in the middle of the year, a reduction that cannot be definitively attributed to reduced mortality overall, given it coincides with a quiescent period in the reproductive cycle where birds spend less time ashore [57] and monitoring intensity declines. Submissions in November and December are primarily neonatal hoiho, a cohort that would arguably be predated or otherwise go undetected were it not for the intensive nest site monitoring of the Yellow-Eyed Penguin Trust. This could suggest widespread failure in rearing of hoiho neonates, driven by declining diet quality of adult hoiho [58] and supported by evidence of alterations in benthic biodiversity due to changing climate [29] and the impact of commercial fisheries [49]. Alternatively, this may simply be a normal consequence of highly productive breeding grounds. In contrast, no such spike in neonatal mortality is evident within the kororā dataset. However, the intensive research and conservation interest in hoiho since the 1980's, particularly monitoring of breeding sites and nests [29] likely contributes to this discrepancy, and would only be resolved by focused and targeted research of neonates in both species. Further, the lack of a strong correlation identified between any of the signalment variables of species, sex, life stage, and body condition score, at the level of all species and all years, demonstrates the varied and nuanced drivers which influence submission of *Sphenisciformes* to the Wildbase Pathology Register. Ultimately, the distribution and demographics of species seen represents an interplay between research interests, conservation efforts, financial resources, and species biology, layered upon the practicalities of specimen acquisition when efforts are not specifically directed. It is likely this confluence of

factors, possibly alongside others, which has influenced the variation in numbers submitted to the database each year. As previously, whether these structural biases are a concern relates back to the purpose of a wildlife health database, which warrants reflection and refinement in an ongoing manner to ensure parity between the intended scope and outcomes.

Analysis of individual diagnoses extracted from the Wildbase Pathology Register must be considered in light of the relative abundance of hoiho submissions. Of the 135 unique diagnoses encountered, over half (54%) were only encountered once. This may reflect the variety of pressures faced by penguins in New Zealand, with infectious disease a recognised threat in hoiho [28] and predation a recognised threat in kororā [59]. Emaciation is by far the most common individual diagnosis Identified across all species. On the surface this appears to substantiate the deleterious effects of habitat degradation on the diet of penguins in New Zealand [29,58]. However, grouping individual diagnoses according to the DAMNITV pneumonic [60], as has been utilised in similar contexts [44,61], unveils an alternative or contributory mechanistic explanation, with infectious and inflammatory conditions becoming the most common category. This appears contrary to the low relative importance of infectious disease to the conservation of penguins as an Order [28]. The relative abundance of literature on infectious diseases in hoiho, ranging from mass mortality events to diphtheritic stomatitis [37–39], corroborates the importance of infectious and inflammatory disease to conservation of this species [28]. By contrast, traumatic diagnoses are by far the most common broad category in kororā. This is consistent with the wider literature on the threat of predation to kororā both within New Zealand and without [62,63]. In both instances however, the interplay between predation, infectious and inflammatory disease and the environment must be considered, as predation of an individual animal may be secondary to disease [64], and environmental degradation may exacerbate infectious and inflammatory disease presence within a population [50,65]. Regardless, this stark difference between diagnoses in hoiho and kororā, coupled with the proportional dominance of hoiho in the dataset, highlights the need for consistency in reporting signalment and epidemiologic variables so that sub-populations can be easily identified, and such relationships elucidated.

One of the barriers to robust analysis encountered in this review is redundancy in data where contributors are not restricted to specific terms when entering key categorical data. Native within the Wildbase Pathology Register, synonyms and typographical variations of species and subspecies are present as discrete taxonomic categories. Similar redundancies are also encountered for the signalment variables of sex and age classification. This issue of reporting consistency is clearest in the analysis of submitting organisations, where 130 different organisation names were revised into 46 unique organisations due to variations in typography and naming convention. Standardisation of such epidemiologic data has previously been shown to increase the utility of animal disease surveillance databases [51].

It is apparent that standardisation of signalment data may be helpful in increasing the ease of review, arguably the biggest challenge lies in the very nature of clinicopathologic reports which comprise such databases. While there is a long history of the successful implementation of systems to standardise key diagnostics findings from medical reports, for example; the WHO's perpetually updated International Statistical Classification of Diseases and Related Health Problems [66], some authors recognise the potential for data loss with such coding systems [67,68]. In addition, the manual analysis of large volumes of clinical notes likely introduces a level of subjectivity from the person(s) performing the analysis [67]. Across published large-scale reviews of animal morbidity and mortality, manual reading and interpretation of each individual report appears to serve as the basic methodology used [8,10,13]. This time-consuming approach is likely a barrier to regular review of such databases, reducing their utility as early detection surveillance tools, or drivers of

timely management or research actions [19,51]. There is however an evolving body of literature describing innovative approaches, such as text-mining, for information retrieval and interactive dashboards for information display, which may expedite free-text clinical data analysis [26,69].

## Conclusion

Necropsy databases such as the Wildbase Pathology Register contain information which can reveal historical and near contemporary trends in wildlife morbidity and mortality [19], with obvious implications for informing conservation management and policy, as well as biosecurity and public health. However, identification of these trends relies on regular review of the records, and evaluation of the relevance of findings in light of the biases common to such databases [6]. In the case of New Zealand's *Sphenisciformes*, research and conservation interests around hoiho make the database less representative of trends at the order level. However, when data are subset by species it does broadly support the trends noted in wider literature on penguin conservation [63]. Explicitly, data contained within the Wildbase Pathology Register suggests that there is a likely interaction between poor nutrition, trauma, and infectious and inflammatory disease as major contributors to the morbidity and mortality of New Zealand penguins. We propose that the dashboard-stye reporting and text-mining methodologies found in electronic clinical data analysis more broadly [67,70–72], are applied to interrogate and communicate key findings from wildlife necropsy datasets, promoting more direct and timely reviews. However, to achieve this, existing barriers such as funding to employ personnel with expertise, as well as issues of data-sharing and intellectual property, will need to be resolved.

## Acknowledgements

The authors thank all those who contributed to the accessions within the Wildbase Pathology Register. Ranging from veterinarians performing the post-mortems, to the Conservationists, lay or professional, that facilitate the identification, collection, and transport of specimens, especially members of the Yellow Eye'd Penguin Trust. We also would like to thank Daniel White of Cumhaill Genetic Solutions for assistance with rendering of Fig 3. In undertaking this research on penguins of Aotearoa, particularly hoiho, the authors acknowledge Te Rūnanga o Ngāi Tahu alongside the wider Tangata Whenua and their kaitiakitanga. Finally, thanks must also go to Lyndell Paniora, Kaupapa Māori Advisor for Auckland Zoo | Te Whare Kararehe o Tāmaki Makaurau, for her understanding and advice.

## Author contributions

**Conceptualization:** Stefan Saverimuttu, Kate McInnes, Bethany Jackson.

**Data curation:** Stefan Saverimuttu, Stuart Hunter, Brett Gartrell, Kate McInnes.

**Formal analysis:** Stefan Saverimuttu, Bethany Jackson.

**Investigation:** Stefan Saverimuttu, Bethany Jackson.

**Methodology:** Stefan Saverimuttu, Bethany Jackson.

**Project administration:** Stefan Saverimuttu, An Pas, James Chatterton, Bethany Jackson.

**Resources:** Stuart Hunter, Brett Gartrell, Kate McInnes, Bethany Jackson.

**Supervision:** Stuart Hunter, Kate McInnes, Kristin Warren, An Pas, James Chatterton, Lian Yeap, Bethany Jackson.

**Visualization:** Stefan Saverimuttu.

**Writing – original draft:** Stefan Saverimuttu, Bethany Jackson.

**Writing – review & editing:** Stefan Saverimuttu, Stuart Hunter, Brett Gartrell, Kate McInnes, Kristin Warren, An Pas, James Chatterton, Lian Yeap, Bethany Jackson.

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
