## [Decision Letter · Decision Letter 0]

22 Jan 2025

PONE-D-24-52763What's in a database? Insights from a retrospective review of penguin necropsy records in Aotearoa New ZealandPLOS ONE

Dear Dr. Saverimuttu,

Thank you for submitting your manuscript to PLOS ONE. After careful consideration, we feel that it has merit but does not fully meet PLOS ONE’s publication criteria as it currently stands. Therefore, we invite you to submit a revised version of the manuscript that addresses the points raised during the review process.

We look forward to receiving your revised manuscript.

Kind regards,

Stephen Raverty

Academic Editor

PLOS ONE

Journal Requirements:

3. In the online submission form you indicate that your data is not available for proprietary reasons and have provided a contact point for accessing this data. Please note that your current contact point is a co-author on this manuscript. According to our Data Policy, the contact point must not be an author on the manuscript and must be an institutional contact, ideally not an individual. Please revise your data statement to a non-author institutional point of contact, such as a data access or ethics committee, and send this to us via return email. Please also include contact information for the third party organization, and please include the full citation of where the data can be found.

4. We note that Figure 3 in your submission contain [map/satellite] images which may be copyrighted. All PLOS content is published under the Creative Commons Attribution License (CC BY 4.0), which means that the manuscript, images, and Supporting Information files will be freely available online, and any third party is permitted to access, download, copy, distribute, and use these materials in any way, even commercially, with proper attribution. For these reasons, we cannot publish previously copyrighted maps or satellite images created using proprietary data, such as Google software (Google Maps, Street View, and Earth). For more information, see our copyright guidelines: http://journals.plos.org/plosone/s/licenses-and-copyright.

a. You may seek permission from the original copyright holder of Figure 3to publish the content specifically under the CC BY 4.0 license. 

Additional Editor Comments:

Each reviewer identified merit in the manuscript, it is well organized and the data is detailed. The comments provided by the reviewers will enhance the quality of the manuscript.

Reviewers' comments:

Reviewer's Responses to Questions

**Comments to the Author**

1. Is the manuscript technically sound, and do the data support the conclusions?

Reviewer #1: No

Reviewer #2: Yes

2. Has the statistical analysis been performed appropriately and rigorously? 

Reviewer #1: No

Reviewer #2: Yes

3. Have the authors made all data underlying the findings in their manuscript fully available?

Reviewer #1: Yes

Reviewer #2: Yes

4. Is the manuscript presented in an intelligible fashion and written in standard English?

Reviewer #1: Yes

Reviewer #2: Yes

5. Review Comments to the Author

Reviewer #1: The authors aimed at performing a mix literature review and base description of the Wildbase Pathology Register database in New Zealand, focusing on Sphenisciformes. The paper is very well written and available to wider audiences. However, while I admire the efforts that were put into curating the dataset, I’m left skeptical about the goal of the paper. From my perspective, the manuscript in its current form describes what any researcher does when doing big data analysis (e.g. start with extensive data curation, explore the dataset), but significantly lacks multiple key components, such as a ‘why’ statement and quantitative statistical analysis. At the moment, the paper is mostly descriptive and does not inform on the inferences that can be made from large scale publicly available datasets such as this one. That being said, I think there is value to the work of the authors, and that a few improvements would make the manuscript much stronger and suitable for publication.

First and foremost, I would recommend the authors to strengthen the last part of the introduction. I really enjoyed reading this section, it’s clear and concise, but it lacks purpose toward the end. While the aim and intentions are clearly defined, I do not feel that they align with the rest of the work. For instance, the first sentence of the discussion reads: we describe spatial and temporal trends in submission data for penguins in the Wildbase Pathology Register, noting the influence of variables such as year, month, age, species, and location of submission on the dataset. However, in the introduction, the authors claim that they intend to ‘ predict limitations of demographic representation across species’, but I fail to see any spatial analysis in the rest of the paper, aside from figure 3, which simply describes samples locations. In my opinion, statistically linking human population density from available online raster with the necropsy data from the Wildbase Pathology Register would have been much more interesting and useful than simply describing samples abundance across New Zealand. The same is true for temporal trends, which could be more robustly analyzed than the current descriptive format.

Related to that, I believe that statistical approach in this manuscript to be flawed. When working with such wealth of data, much more informative statistical approaches than p-value difference-based tests should be used. When analysing big datasets like this one, the question should not be: is there a difference? But what is the magnitude of the difference?, at least in my opinion. As such, I would recommend the authors to at least use a generalized mixed effect models (package: lme4) in R to infer on the wide array of factors that they explore.

Lastly, if the goal of the current manuscript is to only explore the dataset and not infer from it, I would recommend the authors to include a figure showing the trend of publications on morbidity/diseases of Sphenisciformes vs what’s being made available by the public. This could help informing on current biases in the datasets (e.g. is the big jump in 2005 related to more reporting?).

Reviewer #2: This manuscript provides a very detailed documentation of the use of retrospective mining of a wildlife disease database using Sphenisciformes, and in particular the hoiho (yellow eyed penguin, Megadyptes antipodes) and kororā (blue penguin, Eudyptula minor). The manuscript is well written and follows a logical approach. Below I have included a couple of minor questions, clarifications, and suggested comments for consideration by the authors that I think will further add to the quality of the manuscript:

1. Abstract: I appreciate that the authors have also listed some important identified limitations right in the abstract as this further highlights the factors that need consideration in such retrospective studies and could help highlight some factors to consider in the generation of these databases going forward.

2. Material and methods; line 140-143; there is some repetition here with the introduction section; however, I think this still fits into the word count recommendations; however, if not it would make most sense to have this more detailed description still under the materials and methods section.

3. Material and methods; line 146-148; I think this would imply that necropsies of deceased wildlife are performed at the Auckland Zoo in addition, but it might be somewhat unclear to some readers, and could it be possible that this could be read that necropsies from captive or Zoo managed breeding programs could be included?

4. Material and methods; line 156-158; was this interpretation conducted by a single person or multiple authors? And how was this standardized?

5. Material and methods; table 1: The depiction of what information needed to be reviewed and interpreted in table form is a very nice descriptive visual. Why the decision to divide by 1.8 specifically? How was this decided?

6. Material and methods; line 172-173; was this ambiguous enough that it could not be included in the data that was interpreted? Would the availability of this information have any additional penguin conservation impact albeit then still even more biased?

7. Material and methods; line 180-190; do the authors touch on potential causes for lack of representation of the other six species of penguins in the database?

8. Material and methods; line 184-186; this makes sense. More a question of curiosity but why so few submissions in the earlies years? Was this impacted by conservation decisions and allocation of government funding? If yes to the latter, it might further allow for illustration of the important impact these reviews or database generation can have on governmental decisions.

9. Material and methods; line 187-192; does spatial location data allow for identification of differences between specific populations if significant site fidelity exists?

10. Material and methods; line 199-201; I assume that part of the redundancy consolidation also included consistency in assigning specific diagnoses to DAMNITV categories and that this was again standardized as referred to in comment 4 above?

11. Results; line 209-215; This is a very clear presentation of the demographics of species submissions and the data in general with both percentage and numbers being reported. Thank you for this clarity.

12. Results; line 221-225; please see comment 5 above with regards to the 1.8 factor.

13. Results; line 236-238; please refer to comment 8 why these two temporal spikes (2005 and 2008) in submissions? And how does that affect data interpretations? This is partially explained for the 2011 submissions of korora with the Rena oil spill.

14. Results; line 305-308; Could this conversely also be affected by spatial association to the nearest veterinary diagnostic laboratory and ease of transport/accessibility to these sites? E.g. majority of inland veterinary submissions originated from Palmerston North. Is this something that could also be further interpreted from data sets such as this to increase accessibility of diagnostic testing/evaluation for these wildlife species - e.g. penguins could be relatively easily couriered?

15. Results; line 322; is trauma related to marine traffic or land traffic? Is it worth further separating these out?

6. PLOS authors have the option to publish the peer review history of their article (what does this mean? ). If published, this will include your full peer review and any attached files.

**Do you want your identity to be public for this peer review?** For information about this choice, including consent withdrawal, please see our Privacy Policy .

Reviewer #1: No

Reviewer #2: **Yes: ** Heindrich N Snyman

---

## [Author Response · Author response to Decision Letter 1]

11 Mar 2025

Response to specific reviewer comments has been uploaded as a separate document however, also placed here for convivence.

Response to Reviewers

Responses to the Academic Editor

We have reviewed these documents and made some adjustments. Headings have now been accurately altered to fit these guidelines, author affiliations symbols altered, figure and table descriptions brought in line.

Please see data availability statement which details restrictions around public availability of this data alongside how access to the data can be provided for legitimate scientific interests and how this access will be maintained into the future. These details are provided in accordance with previous email correspondence with PLOS One.

3. In the online submission form you indicate that your data is not available for proprietary reasons and have provided a contact point for accessing this data. Please note that your current contact point is a co-author on this manuscript. According to our Data Policy, the contact point must not be an author on the manuscript and must be an institutional contact, ideally not an individual. Please revise your data statement to a non-author institutional point of contact, such as a data access or ethics committee, and send this to us via return email. Please also include contact information for the third party organization, and please include the full citation of where the data can be found.

Three contacts, across two institutions are provided within the Data Availability Statement according to previous email correspondence with PLOS One.

4. We note that Figure 3 in your submission contain [map/satellite] images which may be copyrighted. All PLOS content is published under the Creative Commons Attribution License (CC BY 4.0), which means that the manuscript, images, and Supporting Information files will be freely available online, and any third party is permitted to access, download, copy, distribute, and use these materials in any way, even commercially, with proper attribution. For these reasons, we cannot publish previously copyrighted maps or satellite images created using proprietary data, such as Google software (Google Maps, Street View, and Earth). For more information, see our copyright guidelines: http://journals.plos.org/plosone/s/licenses-and-copyright.

a. You may seek permission from the original copyright holder of Figure 3to publish the content specifically under the CC BY 4.0 license.

We have altered the Methods, relevant figure caption, and reference list to highlight that shapefiles used for this map were obtained under Creative Commons Attribution 4.0 International. This is also reflected within the Data Availability Statement.

Responses to Reviewer 1

Reviewer #1: The authors aimed at performing a mix literature review and base description of the Wildbase Pathology Register database in New Zealand, focusing on Sphenisciformes. The paper is very well written and available to wider audiences. However, while I admire the efforts that were put into curating the dataset, I’m left skeptical about the goal of the paper. From my perspective, the manuscript in its current form describes what any researcher does when doing big data analysis (e.g. start with extensive data curation, explore the dataset), but significantly lacks multiple key components, such as a ‘why’ statement and quantitative statistical analysis. At the moment, the paper is mostly descriptive and does not inform on the inferences that can be made from large scale publicly available datasets such as this one. That being said, I think there is value to the work of the authors, and that a few improvements would make the manuscript much stronger and suitable for publication.

First and foremost, I would recommend the authors to strengthen the last part of the introduction. I really enjoyed reading this section, it’s clear and concise, but it lacks purpose toward the end. While the aim and intentions are clearly defined, I do not feel that they align with the rest of the work. For instance, the first sentence of the discussion reads: we describe spatial and temporal trends in submission data for penguins in the Wildbase Pathology Register, noting the influence of variables such as year, month, age, species, and location of submission on the dataset. However, in the introduction, the authors claim that they intend to ‘ predict limitations of demographic representation across species’, but I fail to see any spatial analysis in the rest of the paper, aside from figure 3, which simply describes samples locations. In my opinion, statistically linking human population density from available online raster with the necropsy data from the Wildbase Pathology Register would have been much more interesting and useful than simply describing samples abundance across New Zealand. The same is true for temporal trends, which could be more robustly analyzed than the current descriptive format.

We have altered the wording of the described portion of the introduction clarify the aims of the study such that they align better with the overall scope of the manuscript. Further, more discussion has been added around the results of the spatiotemporal analysis in what is now the 3rd paragraph of the Discussion. We appreciate this was a gap within the discussion and appreciate this reviewers comments in this regard.

Related to that, I believe that statistical approach in this manuscript to be flawed. When working with such wealth of data, much more informative statistical approaches than p-value difference-based tests should be used. When analysing big datasets like this one, the question should not be: is there a difference? But what is the magnitude of the difference?, at least in my opinion. As such, I would recommend the authors to at least use a generalized mixed effect models (package: lme4) in R to infer on the wide array of factors that they explore.

Lastly, if the goal of the current manuscript is to only explore the dataset and not infer from it, I would recommend the authors to include a figure showing the trend of publications on morbidity/diseases of Sphenisciformes vs what’s being made available by the public. This could help informing on current biases in the datasets (e.g. is the big jump in 2005 related to more reporting?).

Whilst we appreciate the reviewer’s viewpoint, and can see where it is coming from, we purposefully approached the statistics in a conservative and largely descriptive way, owing to the underlying data generating mechanism of our dataset, which leads to a highly biased dataset (as described) that would make a more holistic analysis such as GLMM inappropriate in our view. We discussed this with colleagues familiar with the statistical approach (GLMM), and our data, and were discouraged from pursuing this model. We feel our discussion highlights this, and that a descriptive approach is more suited than making overarching inferences from data that is driven by disparate underlying mechanisms.

Responses to Reviewer 2

Reviewer #2: This manuscript provides a very detailed documentation of the use of retrospective mining of a wildlife disease database using Sphenisciformes, and in particular the hoiho (yellow eyed penguin, Megadyptes antipodes) and kororā (blue penguin, Eudyptula minor). The manuscript is well written and follows a logical approach. Below I have included a couple of minor questions, clarifications, and suggested comments for consideration by the authors that I think will further add to the quality of the manuscript:

1. Abstract: I appreciate that the authors have also listed some important identified limitations right in the abstract as this further highlights the factors that need consideration in such retrospective studies and could help highlight some factors to consider in the generation of these databases going forward.

We appreciate the reviewers comments as this became apparent within our data and so was intentionally highlighted within the abstract.

2. Material and methods; line 140-143; there is some repetition here with the introduction section; however, I think this still fits into the word count recommendations; however, if not it would make most sense to have this more detailed description still under the materials and methods section.

Wording in the Introduction and Methods sections have been amended in line with this observation. We appreciate that modifications such as this are vital to ensure manuscripts are presented as concisely as possible.

3. Material and methods; line 146-148; I think this would imply that necropsies of deceased wildlife are performed at the Auckland Zoo in addition, but it might be somewhat unclear to some readers, and could it be possible that this could be read that necropsies from captive or Zoo managed breeding programs could be included?

The wording of these lines has been altered to remove this well observed ambiguity.

4. Material and methods; line 156-158; was this interpretation conducted by a single person or multiple authors? And how was this standardized?

Have clarified this fact within the Methods. Interpretation was standardised by it being conducted by a single observer (the primary author)

5. Material and methods; table 1: The depiction of what information needed to be reviewed and interpreted in table form is a very nice descriptive visual. Why the decision to divide by 1.8 specifically? How was this decided?

1.8 was chosen as this is the ratio needed to convert a 1-9 scale to a 1-5 scale as close as is practical. This has now been clarified within the Methods.

6. Material and methods; line 172-173; was this ambiguous enough that it could not be included in the data that was interpreted? Would the availability of this information have any additional penguin conservation impact albeit then still even more biased?

Reporting of proximate cause of death was often ambiguous as suggested, however the primary reason it was not included was that ‘final diagnoses’ were often reported as a simple list with no formal rank of proximity to death.

7. Material and methods; line 180-190; do the authors touch on potential causes for lack of representation of the other six species of penguins in the database?

Further discussion on this point has been added in paragraph 4 of the discussion

8. Material and methods; line 184-186; this makes sense. More a question of curiosity but why so few submissions in the earlies years? Was this impacted by conservation decisions and allocation of government funding? If yes to the latter, it might further allow for illustration of the important impact these reviews or database generation can have on governmental decisions.

We explored this with relevant stakeholders with whom we had direct contact. Your inference is largely what is thought to have happened, however we decided to not directly broach this subject within the manuscript as it is largely speculation with no concrete associated documentation to refer to. In lieu of direct discussion, lines 424 -428 of the Discussion now speak to this with reference to the broader findings of this study.

9. Material and methods; line 187-192; does spatial location data allow for identification of differences between specific populations if significant site fidelity exists?

Some site fidelity for separate populations of these species does indeed exist. However, the inconsistency of location data reporting combined with steps already taken which involved combination and interpretation of existing data (see lines 187 -189) makes us reluctant to use this data to draw firmer conclusions such as those described here. Additionally, the influence of sites where medical treatment is provided to wildlife within this dataset would complicate such analysis.

10. Material and methods; line 199-201; I assume that part of the redundancy consolidation also included consistency in assigning specific diagnoses to DAMNITV categories and that this was again standardized as referred to in comment 4 above?

Indeed. The DAMNITV classifications were applied to the unique diagnoses themselves which were then mapped back to the individual accessions, rather than assigning the classifications on a per accession basis. Wording within the Methods has been altered to clarify this (lines 197-202)

11. Results; line 209-215; This is a very clear presentation of the demographics of species submissions and the data in general with both percentage and numbers being reported. Thank you for this clarity.

Thank you.

12. Results; line 221-225; please see comment 5 above with regards to the 1.8 factor.

Response to Comment 5 applies here: 1.8 was chosen as this is the ratio needed to convert a 1-9 scale to a 1-5 scale as close as is practical. This has now been clarified within the Methods.

13. Results; line 236-238; please refer to comment 8 why these two temporal spikes (2005 and 2008) in submissions? And how does that affect data interpretations? This is partially explained for the 2011 submissions of korora with the Rena oil spill.

See Response to comment 8

14. Results; line 305-308; Could this conversely also be affected by spatial association to the nearest veterinary d

---

## [Editor Report · Decision Letter 1]

14 Mar 2025

What's in a database? Insights from a retrospective review of penguin necropsy records in Aotearoa New Zealand

PONE-D-24-52763R1

Dear Dr. Saverimuttu,

We’re pleased to inform you that your manuscript has been judged scientifically suitable for publication and will be formally accepted for publication once it meets all outstanding technical requirements.

Kind regards,

Stephen Raverty

Academic Editor

PLOS ONE

Additional Editor Comments (optional):

Thank you for your detailed responses to the reviewers suggestions and comments. The content and flow of the paper is improved. The paper is a valuable contribution to the health of penguins in Aotearoa New Zealand.
---

## [Editor Report · Acceptance letter]

PONE-D-24-52763R1

PLOS ONE

Dear Dr. Saverimuttu,

I'm pleased to inform you that your manuscript has been deemed suitable for publication in PLOS ONE. Congratulations! Your manuscript is now being handed over to our production team.

Kind regards,

on behalf of

Dr. Stephen Raverty

Academic Editor

PLOS ONE